# Synergetic binary organocatalyzed ring opening polymerization for the precision synthesis of polysiloxanes
Hiroshi Okamoto[1], Atsushi Sogabe[1] & Satoshi Honda [ORCID][2] ✉

Organocatalytic ring-opening polymerization (ROP) is a versatile method for synthesizing well-defined polymers with controlled molecular weights, dispersities, and nonlinear macromolecular architectures. Despite spectacular advances in organocatalytic ROP, precision synthesis of polysiloxanes remains challenging due to the mismatch in polarity between highly polar initiators and nonpolar monomers and polymers and the difficulty in suppressing the formation of scrambling products via transetherification reactions during ROP of cyclic siloxanes. Here, we describe a binary organocatalytic ROP (BOROP) of hexamethylcyclotrisiloxane (D3) employing organic bases as catalysts and (thio)ureas as cocatalysts. The BOROP of D3 using triazabicyclodecene (TBD) and (thio) ureas generates polydimethylsiloxanes (PDMSs) with narrow dispersity ($M_w/M_n < 1.1$). Despite the similar basicities of TBD and 1,8-bis(tetramethylguanidino)naphthalene (TMGN), which is known as a proton sponge, a unitary organocatalytic system using TMGN was inactive for the ROP of D3. When the TMGN was paired with acidic urea, the BOROP of D3 yielded PDMSs with narrow dispersity ($M_w/M_n < 1.1$). Data suggest that the synergetic effect of TMGN and urea is results in an unprecedented activation–deactivation equilibrium between dormant and propagating species. The benefits of the present BOROP system are demonstrated by the formation of PDMS elastomers with more uniform network structures that are highly stretchy and have excellent mechanical properties.

Ring-opening polymerization (ROP) is an important class of methodologies for synthesizing polymers with controlled molecular weights (MWs) and narrow dispersities (Đs)[1]. Encouraged by the first report of organocatalytic ROP utilizing 4-dimethylaminopyridine (DMAP) by Hedrick et al. in 2001[2], herculean efforts over the years in this area have led to the development of a great number of organocatalysts such as amidine and guanidine bases[3,4], N-heterocyclic carbenes (NHCs)[5,6], phosphazenes[7,8], and urea anions[9]. These catalysts have been shown to exhibit excellent activity, selectivity, and compatibility with various monomers including epoxides, lactones, and carbonates, to mention a few, in addition to often being considered less toxic than metal-based catalysts[10–12].

Polysiloxanes are industrially important materials that cannot be replaced by carbon-based polymers due to their unique properties such as high thermal stability, low surface energy, and excellent biocompatibility[13]. Historically, polysiloxanes have been synthesized by equilibrium polymerization using strong acids or catalyst-free ROP of cyclic siloxanes[14,15]. As polysiloxanes synthesized by equilibrium polymerization have broad

dispersity ($M_w/M_n = Đ$), synthesizing polysiloxanes by ROP of cyclic siloxanes is important when expected applications require controlled MW with narrow Đ[16,17]. In particular, the use of organic catalysts in the ROP of cyclic siloxanes is attractive as reported by Hedrick, Waymouth, and coworkers[18,19]. Unlike other organic catalysts, ROP of 2,2,5,5-tetramethyl-1-oxa-2,5-disilacyclopentane (TMOSC) using 1,5,7-triazabicyclo[4.4.0]dec-5-ene (TBD) after full conversion of monomers did not show a broadening in dispersity[18]. They also reported that organocatalytic ROP of cyclic siloxanes initiated from silanols[18] can avoid the inclusion of labile Si-O-C linkages, or silylether to the resulting polysiloxane. Inspired by these prominent studies, we synthesized star-shaped polydimethylsiloxanes (PDMSs) with controlled MWs and Đs upon organocatalytic ROP of hexamethylcyclo-trisiloxane (D3) initiated from trifunctional silanols[20]. Although a mixture of TBD and trifunctional silanol showed poor solubility in solvents for polymerization due to the mismatch of the highly polar nature of initiating system with nonpolar monomers and polymers, an initiating system using urea anions overcomes this problem. In fact, ROP using urea anions affords

---

[1]MIRAI Technology Institute, Shiseido Co. Ltd, 1-2-11 Takashima, Nishi-ku, Yokohama, Kanagawa 220-0011, Japan. [2]Graduate School of Arts and Sciences, The University of Tokyo, 3-8-1 Komaba, Meguro-ku, Tokyo 153-8902, Japan. ✉e-mail: c-honda@mail.ecc.u-tokyo.ac.jp

polymers with a narrow Đ when the conversion of D3 is low. However, with urea anions, the ROP reaches equilibrium when the conversion of D3 exceeds approximately 50%, and the Đ of the resulting PDMS in the mixture broadens[20]. Hence, in addition to solubilizing the initiating system, developing an efficient catalyst system that can avoid equilibrium polymerization with the ROP of siloxanes[21] remains a crucial challenge.

Herein, we report the binary organocatalytic ROP (BOROP) of D3 using organic bases as catalysts and (thio)ureas as cocatalysts. The synergetic use of TBD and (thio)urea for the ROP of D3 enables both the solubilization of multifunctional silanols in THF and the production of PDMS with a narrow Đ until the monomer conversion reaches approximately 90%. Moreover, we investigated the applicability of proton sponges for the ROP of D3. While a unitary organocatalytic system using proton sponges was inactive, ROP of D3 proceeded with the synergetic use of 1,8-bis(tetramethylguanidino)naphthalene (TMGN) and urea. In the cocatalytic system with TBD and urea, ROP is accelerated with less acidic urea. Despite the similar basicities of TBD and TMGN, ROP was accelerated by the addition of more acidic urea. Mechanistically, this suggests that the BOROP system using TMGN is in activation–deactivation equilibrium between dormant and propagating species, similar to what has been reported for living radical polymerization. The advantage of the present BOROP system was finally confirmed by the excellent mechanical properties of highly stretchy elastomeric PDMS formed from three-armed star-shaped PDMSs. The developed BOROP would be highly beneficial for the precision synthesis of polysiloxanes and enables facile access to those with branched macromolecular architectures with narrow Đ.

## Results and discussion
### ROP of D3 with a unitary organocatalytic system

We first surveyed the basicity of a series of organic bases. The $pK_a$ values of the conjugate acids of these bases ($pK_{BH}^+$) in MeCN indicate similar catalytic activities between DMAP ($pK_{BH}^+ = 17.95$)[22] and 1,8-bis(dimethylamino)naphthalene (DMAN) ($pK_{BH}^+ = 18.18$)[23] and between TBD ($pK_{BH}^+ = 26.03$)[22] and TMGN ($pK_{BH}^+ = 25.10$)[23] (Fig. 1). The ROP of D3 in THF ([D3] = 2.4 M) initiated from trifunctional silanol ($I_3$) (Fig. 2a) with these organic bases readily revealed that less basic DMAP and DMAN, which are known as a proton sponge, are inactive under the polymerization conditions tested (Table 1, entries 1 and 2). Similar to the pioneering study reported by Waymouth, Hedrick, and coworkers[18], the ROP of D3 with strongly basic TBD proceeded, and the conversion reached 92% with a polymerization time of 120 min (Table 1, entry 3). Plots of monomer conversion against time determined using size exclusion chromatography (SEC) showed a monotonic increase in conversion, whereas Đ apparently broadened after 60 min (Fig. 2b). While a previous study on the ROP of TMOSC indicated that TBD could prevent the formation of scrambling products via transetherification reactions[19], in the case of D3, the considerably smaller $M_n$ than that calculated from conversion ($M_{n,theo}$) and the broadening of Đ at higher conversions suggest the formation of oligomers

and scrambling products (Fig. 2c). Considering that the $pK_{BH}^+$ values of TBD and TMGN are comparable (Fig. 1), in terms of basicity, the ROP of D3 using TMGN as an organocatalyst should proceed. However, in prior literature, the organocatalytic ROP of octamethylcyclotetrasiloxane using TMGN was performed at 65 °C, suggesting difficulty in controlling ROP with TMGN[24]. In fact, when ROP of D3 was attempted with TMGN, no monomer consumption was observed at 25 °C (Table 1, entry 4). This suggests that basicity is not the only factor affecting catalytic activity in the ROP of D3, and the bifunctionality of TBD[25], characterized by its dual action as both a hydrogen-bonding donor and acceptor, is probably responsible for its reactivity. Compared to TBD, TMGN is more sterically hindered; thus, the difference in the steric configuration of the basic reactive center may be responsible for the unitary polymerizability.

### ROP of D3 with binary organocatalytic system

To examine the effect of cocatalysts on the ROP of D3 and to test its generality, we screened a series of ureas to be paired with TBD (Fig. 3a). We first monitored the interactions among $I_3$, TBD, and U(Cy) in THF based on deuterated solvent-free benchtop [1]H NMR (80 MHz) spectrometry. While signals that should appear at 1–4 ppm in the [1]H NMR spectrum overlap with THF-derived signals, the signals appearing in the spectrum of $I_3$ (Supplementary Fig. 1a, marked a, b, and c) corresponded to those appearing in the separately measured [1]H NMR (500 MHz) spectrum (Supplementary [1]H NMR). and U(Cy) (Supplementary Fig. 1b, marked $Ph_U$), which corresponded to their reported values and that appeared in the spectrum of their mixture but remained unchanged (Supplementary Fig. 1c). When TBD was added to the THF mixture of $I_3$, the resulting mixture became turbid due to the insolubility of the silanol-derived ionic species in THF. The [1]H NMR spectrum of the soluble fraction showed the disappearance of the signal derived from silanol, and the signal derived from dimethyl protons at 0.23 ppm broadened and split into two signals (marked b), also indicating its poor solubility in THF. (Supplementary Fig. 1d). Furthermore, the signal derived from the benzene ring significantly broadened and its intensity decreased. This decrease in the mobility of protons suggested that the components thought to be dissolved in THF might aggregate. In contrast, while the addition of U(Cy) to the mixture of $I_3$ and TBD significantly increased the intensity of the signal derived from the benzene ring of $I_3$ (marked a), that derived from silanol was still absent (Supplementary Fig. 1e). A more noticeable change is that the split and broadened signals derived from dimethyl groups become one narrowed signal. In addition to the fact that the addition of U(Cy) apparently solubilized the mixture of $I_3$ and TBD, these results also suggest a rapid and reversible exchange reaction between the silanols and TBD/U(Cy). The interactions among $I_3$, TBD, and U(Cy) were further examined based on infrared (IR) spectroscopy. While the IR spectra of $I_3$ and U(Cy) showed broad absorptions derived from O–H (3270 cm$^{-1}$) and N–H (3320 cm$^{-1}$) stretching vibrations (Supplementary Fig. 2a, b), respectively, mixing $I_3$, TBD, and U(Cy) resulted in significant broadening of the absorption at the

**Fig. 1 | Organocatalysts and their $pK_a$s.** Chemical structures of DMAP, DMAN, TBD, and TMGN and their reported $pK_a$s in MeCN.

$pK_{BH}^+$ (MeCN) = 17.95

DMAP

$pK_{BH}^+$ (MeCN) = 18.62

DMAN

$pK_{BH}^+$ (MeCN) = 26.03

TBD

$pK_{BH}^+$ (MeCN) = 25.1

TMGN

**Fig. 2 | Unitary organocatalytic ROP for the synthesis of star-shaped PDMSs. a** ROP of D3 with TBD initiated from $I_3$. **b** Plots of conversion and Đ against time. **c** SEC traces of the products after a series of polymerization time (10, 30, 60, 90, 120 min). Source data for Fig. 2b is available (Supplementary Data 1).

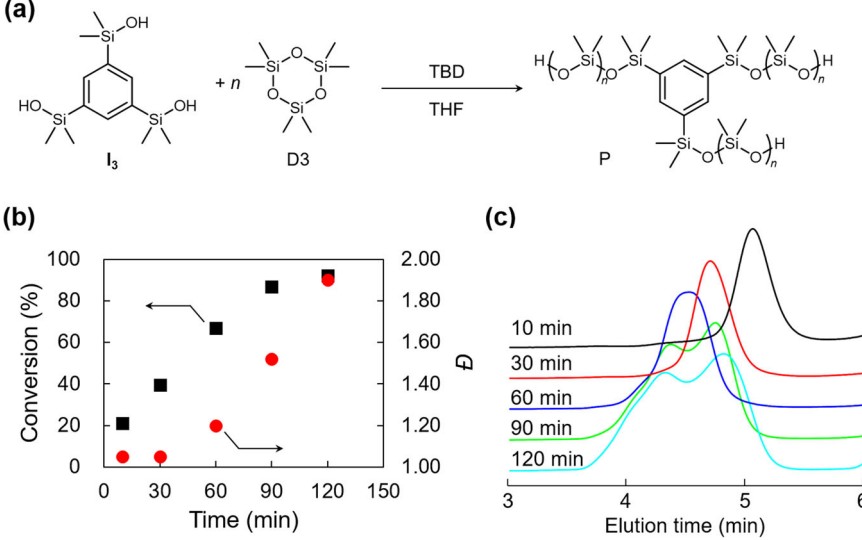

relevant wavelengths (2000–3500 cm⁻¹). This suggests the presence of various types of H-bonding in the mixture and that the H-bonding nature of U(Cy) contributes to the solubilization of $I_3$ in the reaction mixture.

Having confirmed the effective formation of H-bonds between the silanol $I_3$ and TBD/U(Cy), we tested a total of 8 entries of the binary organocatalytic ROP (BOROP) of D3 under different conditions to obtain star-shaped PDMSs with narrow Đ, and the results of the screening are presented in Table 2. Since TBD should extract a proton from urea in the reaction system, the reactivity of BOROP should be of the same order as the reactivity of the urea anion. With that, we started from the use of U(Cy) as a cocatalyst, which has been reported to produce the most reactive urea anion, and readily recognized that the Đ of the product with monomer conversion similar to that of the unitary system (Table 1, entry 3) ($M_n$ = 15900, Đ = 1.90) was markedly narrowed (Table 2, entry 1) ($M_n$ = 35500, Đ = 1.27). A decrease in the amount of TBD further narrowed Đ ($M_n$ = 23400, Đ = 1.16), and a decrease in the amount of U(Cy) resulted in the broadening of Đ ($M_n$ = 24700, Đ = 1.42). These results directly demonstrated the role of urea in narrowing Đ. Next, the effect of the number of CF₃ groups on BOROP was examined. As expected, the increase in the number of CF₃ groups slowed the polymerization (Table 2, entries 4–6). However, it was more interesting that Đ became noticeably narrower as CF₃ increased. Remarkably, the Đ of the PDMS in the polymerization mixture was less than 1.1 even when the monomer conversion reached 99% (Table 2, entry 6). The use of a more acidic TU also afforded PDMS with a narrow Đ but required significantly longer reaction times (Table 2, entries 7 and 8). From these results, the synergetic use of TBD and U(4CF₃) optimizes the BOROP of D3.

### Table 1 | ROP of D3 initiated from $I_3$ with a series of organic bases[a]

| Entry | Cat. | Time (min) | Conv. (%)[b] | $M_n$[c] | Đ[d] |
|---|---|---|---|---|---|
| 1 | DMAP | 120 | 0 | N.D. | N.D. |
| 2 | DMAN | 120 | 0 | N.D. | N.D. |
| 3 | TBD | 120 | 92 | 15,900 (27,600) | 1.90 |
| 4 | TMGN | 120 | 0 | N.D. | N.D. |

[a]The polymerizations were performed at 25 °C. $[I_3]_0$ : [Cat.]₀ : [D3]₀ = 1:1.5:135. $[I_3]_0$ = 0.018 M, [D3]₀ = 2.14 M.
[b]Conversion determined by ¹H NMR.
[c]Number average molecular weight, determined by SEC with RI detector. The number in the parenthesis represents molecular weight calculated from conversion.
[d]Dispersity (Đ = $M_w/M_n$), determined by SEC. SEC measurements were performed after terminating with benzoic acid.

### BOROP of D3 in activation–deactivation equilibrium between dormant and propagating species

Having optimized the BOROP conditions, we next screened organocatalysts (Fig. 1) to be paired with U(4CF₃). While less basic DMAP and DMAN were again inactive even in the presence of U(4CF₃) (Table 3, entries 1 and 2), interestingly, the BOROP of D3 with the synergetic use of TMGN and U(4CF₃) produced PDMSs with narrow Đ (Table 3, entries 3 and 4). The plots of conversion against time showed exceptional linearity (Fig. 4a), and Đ remained narrow even with a polymerization time of 360 min. In the case of the TBD/urea catalytic system, BOROP of D3 should proceed via neutral and imidate-mediated H-bonding mechanisms as reported previously (Fig. 4c)[26]. However, considering that the unitary catalytic system using TMGN was inactive, activation–deactivation equilibrium, similar to that observed for living radical polymerization, such as atom transfer radical polymerization (ATRP)[27], can be proposed for the present synergetic BOROP system using TMGN and U(4CF₃) (Fig. 4d). Thus, when protons derived from silanols are abstracted by TMGN, the formed complex cannot attack the monomer due to the steric hindrance of the methyl groups surrounding the TMGN, as recognized in the unitary ROP system (Table 1, entry 4). On the other hand, if the abstraction of protons derived from U(4CF₃) is in equilibrium, the growing species can attack the monomer (Fig. 4d). Given that the pKa of $I_3$ can be approximately between that of Et₃SiOH (pKa = 13.63) and that of Ph₃SiOH (pKa = 16.57–16.63)[28,29], and that the reported pKa of U(4CF₃) is 13.8[30], it is reasonable to assume this equilibrium. With this activation–deactivation equilibrium between dormant and propagating species, BOROP with less acidic U(Cy) (pKa = 22.8 (25.1))[31] should not proceed because no proton abstraction could occur. This hypothesis was readily supported by the inactivity of the TMGN/U(Cy) cocatalytic system (Table 3, entry 5). The urea anion formed from less acidic urea is more active in the conventional ROP (Fig. 4c), following which the activation–inactivation process (Fig. 4d) mechanistically reverses the relationship between the acidity of urea and the polymerization rate, despite TBD (p$K_{BH}^+$ = 26.03) and TMGN (p$K_{BH}^+$ = 25.10) having similar basicities (Fig. 1). In the present case, the moderate acidity of U(4CF₃) would push the equilibrium to the left-hand side (Fig. 4d), thus resulting in slower polymerization than in the TBD/U(4CF₃) cocatalytic system. The plausibility of this hypothesis was further examined based on IR spectroscopy. First, the proton abstraction from $I_3$ by TMGN was supported by the comparison of IR spectra among $I_3$ (Supplementary Fig. 2a), TMGN (Supplementary Fig. 3a), and their mixture (Supplementary Fig. 3b) by the disappearance of absorption derived from the O–H stretching vibration (3270 cm⁻¹) visible for $I_3$ from their mixture. On the other hand, a comparison of the IR spectra of TMGN (Supplementary Fig. 3a), U(4CF₃) (Supplementary Fig. 3c), and

**Fig. 3 | Binary organocatalytic ROP for the synthesis of star-shaped PDMSs. a** Chemical structures of TBD and cocatalysts. **b** Plots of conversion and Đ against time and (**c**) SEC traces of the products after a series of polymerization time (10, 30, 60, 90, 120 min) (Table 2, entry 1). Source data for Fig. 3b is available (Supplementary Data 1).

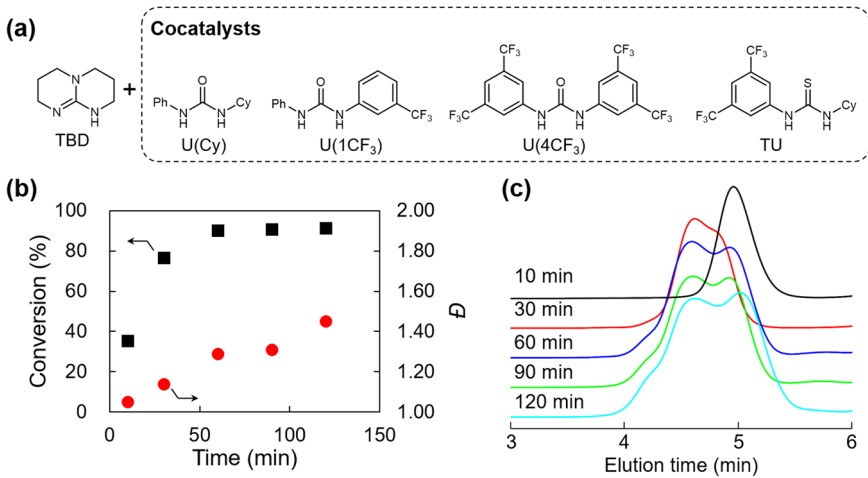

their mixture (Supplementary Fig. 3d) revealed a decrease in the absorption derived from the N–H stretching vibration at approximately 3300 cm$^{-1}$ from their mixture, although the absorption derived from the C=O stretching vibration at 1720 cm$^{-1}$ remained unchanged. If one of the NH protons in U(4CF$_3$) is abstracted by TMGN, the absorption attributed to the C=O stretching vibration should disappear or decrease. Hence, the steric hindrance of TMGN around basic sites likely discourages proton abstraction even from relatively acidic U(4CF$_3$). Furthermore, the IR spectra of the mixture of I$_3$, U(4CF$_3$), and TMGN (Supplementary Fig. 3e) were almost identical to those of the mixture of U(4CF$_3$) and TMGN (Supplementary Fig. 3d). This is consistent with the hypothesized left-biased equilibrium in Fig. 4d. Although the equilibrium should be biased to the left, it is difficult to show the existence of propagating species on the right side of the equilibrium; the fact that the unitary ROP by TMGN did not proceed and that the BOROP by using TMGN and U(4CF$_3$) did is one of the results supporting the existence of the hypothesized equilibrium.

**Fabrication of a highly transparent, elastic, and stretchy elastomer enabled by precisely-defined star-shaped PDMS**

The established BOROP provides unexampled opportunities for various silicone-based materials. To ensure the feasibility of the study, we examined the potential application of our star-shaped PDMSs enabled by BOROP for silicone potting compounds (SPCs). SPCs are an important class of curable

liquid polysiloxanes used, for example, in electronics, medical applications, and microfluidics. Silicone elastomers can be formed simply by mixing two liquid polysiloxanes. To test the applicability of our chemistry to SPC, we synthesized a series of 3-armed star-shaped PDMSs with hydrosilane (**P$_H$**s: **P$_{H(10k)}$**, **P$_{H(25k)}$**, and **P$_{H(27k)}$**) and vinyl end groups (**P$_V$**s: **P$_{V(10k)}$**, and **P$_{V(25k)}$**) based on the developed BOROP of D3 (Supplementary Table 1). The BOROP conditions were controlled to have MWs of approximately 10 kDa and 25 kDa, and these MWs were adjusted to be distinctly lower or slightly higher than the critical entanglement MW ($M_e$) of PDMS (24.5 kDa)[32]; thus these conditions are suitable for examining the effect of MW and entanglement on mechanical properties. The synthesized **P$_H$**s and **P$_V$**s were then mixed in the presence of Karstedt's catalyst so that the hydrosilane and vinyl end groups of **P$_H$**s and **P$_V$**s, respectively, were equimolar, and their behavior as SPCs was examined (Fig. 5a). Moreover, to examine the effect of Đ on the mechanical properties, industrial-grade PDMS with vinyl end groups and a broad Đ (**P$_{VS-7}$**) produced by equilibrium polymerization, such that the relationship between the branching point and the number of end groups is equivalent to that of 3-armed star-shaped PDMS, was also coupled to **P$_H$**s to investigate its behavior as an SPC. Eventually, we fabricated a total of 5 silicon elastomers (**S$_E$**s: **S$_{E(10k,10k)}$**, **S$_{E(10k,VS-7)}$**, **S$_{E(25k,25k)}$**, **S$_{E(27k,10k)}$**, and **S$_{E(10k,VS-7)}$**) from a series of **P$_H$**s and **P$_V$**s. According to the established protocol[33,34], we first evaluated the degree of crosslinking based on IR spectroscopy by the change in the absorbance derived from the Si–H bending vibration. It is clear from the comparison of IR spectra between **P$_{V(10k)}$** (Supplementary Fig. 4a) and **P$_{H(10k)}$** (Supplementary Fig. 4b) that the absorption derived from the Si–H group at 912 cm$^{-1}$ is distinguishable from the other absorptions, and the Si–H absorption also appeared in the mixture of **P$_{V(10k)}$** and **P$_{H(10k)}$** (Supplementary Fig. 4c). In contrast, the Si–H absorption band disappeared from the IR spectrum of **S$_{E(10k,10k)}$** (Supplementary Fig. 4d), indicating the quantitative conversion of the hydrosilane end groups of **P$_{H(10k)}$**. This spectral change is more evident from the comparison of their magnified spectra (Supplementary Fig. 4e). Moreover, the disappearance of Si–H absorption was commonly observed in the spectra of **S$_{E(25k,25k)}$** (Supplementary Fig. 5a), **S$_{E(27k,10k)}$** (Supplementary Fig. 5b), **S$_{E(10k,VS-7)}$** (Supplementary Fig. 5c), and **S$_{E(27k,VS-7)}$** (Supplementary Fig. 5d). All of these results suggest that the degree of crosslinking can be determined to be approximately 100% based on IR spectroscopy.

Next, the rheological properties upon curing a series of mixtures of **P$_H$**s and **P$_V$**s were investigated. The time-dependent plots of $G'$ and $G''$ measured immediately after preparing a series of mixtures commonly showed a significant increase in $G'$ after the induction period (Fig. 5b). After approximately 1500 s, $G'$ reached equilibrium, and $G'$ of the resulting **S$_E$**s varied depending on the starting **P$_H$**s and **P$_V$**s. (Fig. 5b). The equilibrated $G'$ reached after at least two hours ($G_e'$) was determined to be 28 kPa for

**Table 2 | BOROP of D3 initiated from I$_3$ catalyzed by TBD with a series of cocatalysts[a]**

| Entry | Cocatalyst | [I$_3$]$_0$ : [TBD]$_0$ : [HBA]$_0$ : [D3]$_0$ | Time (min) | Conv. (%)[b] | $M_n$[c] | Đ[d] |
|---|---|---|---|---|---|---|
| 1 | U(Cy) | 1:1.5:3:135 | 90 | 91 | 12000 | 1.31 |
| 2 | U(Cy) | 1:0.75:3:135 | 90 | 88 | 23400 | 1.16 |
| 3 | U(Cy) | 1:0.75:1.5:135 | 90 | 93 | 24700 | 1.42 |
| 4[e] | U(1CF$_3$) | 1:0.75:3:135 | 135 | 75 | 28600 | 1.17 |
| 5[e] | U(4CF$_3$) | 1:0.75:3:135 | 60 | 57 | 28200 | 1.07 |
| 6[e] | U(4CF$_3$) | 1:1.5:3:135 | 120 | > 99 | 35700 | 1.09 |
| 7 | TU | 1:1.5:3:135 | 120 | 44 | 12600 | 1.09 |
| 8 | TU | 1:0.75:3:135 | 120 | 26 | 7300 | 1.06 |

[a]The polymerizations were performed at 30 °C. [D3]$_0$ = 2.14 M. Calculated number average molecular weight at the full consumption of D3 was 30,000.
[b]Conversion determined by $^1$H NMR.
[c]Number average molecular weight, determined by SEC with RI detector.
[d]Dispersity (Đ = $M_w/M_n$), determined by SEC. SEC measurements were performed after terminating with benzoic acid.
[e][D3]$_0$ = 2.25 M.

$S_{E(10k,10k)}$ (Fig. 5b, black line), whereas the $G_e$' of $S_{E(10k,VS-7)}$ was approximately 120 kPa (Fig. 5b, blue line), despite the $M_n$s of the starting $P_{V(10k)}$ and $P_{VS-7}$ being essentially equivalent (Supplementary Table 1). As the $Đ$ of $P_{VS-7}$ is broad (Supplementary Table 1, entry 6), the improvement in $G_e$' with $P_{VS-7}$ could be explained by the occurrence of entanglement owing to the presence of long polymer chains. Moreover, $S_{E(25k,25k)}$ had the highest $G_e$' (Fig. 5b, red line) among the tested combinations of $P_H$s and $P_V$s (Supplementary Table 2, entry 3). Apparently, both $P_{H(25k)}$ and $P_{V(25k)}$ have MWs higher than $M_e$ (Supplementary Table 1, codes 2 and 5), thus suggesting the effective formation of entanglement between polymer chains within the network. To verify the effect of entanglement, we next compared the rheological properties of the combination of $P_{H(27k)}$ and $P_{V(10k)}$. In this case, $G_e$' significantly decreased to 27 kPa (Supplementary Table 2, entry 4). If only the effect of MW on the mechanical properties is considered, $G_e$' should increase with decreasing MW of the precursors, i.e., with decreasing

network size, and the present decrease in $G_e$' is presumably due to the decrease in entanglement. On the other hand, the combination of $P_{H(27k)}$ and $P_{VS-7}$ resulted in a decrease in $G_e$' to 33 kPa (Supplementary Table 2, entry 5), even though $P_{VS-7}$ was expected to undergo entanglement. Although the details are not clear, the greater $G_e$' with $P_{H(10k)}$ (Supplementary Table 2, entry 2) than with $P_{H(27k)}$ (Supplementary Table 2, entry 5) may be due to the smaller network size. In any case, eliminating the effect of either MW or entanglement is difficult, and discussing their effects on physical properties separately is difficult. In contrast, it should be noted that these effects can be discussed separately with PDMSs with narrow $Đ$.

Finally, the elastic nature of $S_{E(25k,25k)}$ was examined based on static mechanical analysis. Thus, a tensile test was conducted on $S_{E(25k,25k)}$, and the tensile modulus ($E$), elongation at break ($\varepsilon_B$), tensile strength ($\sigma_T$), and fracture energy ($\Gamma$) were determined to be 0.41 MPa, 330%, 0.48 MPa, and $1.2 \times 10^4$ J m$^{-2}$, respectively, from the stress–strain curve (Fig. 5c). It would be meaningful to compare the mechanical properties of $S_{E(25k,25k)}$ with those of existing SPCs. Notably, the measured $\Gamma$ ($1.2 \times 10^4$ J m$^{-2}$) of our $S_{E(25k,25k)}$ was 10 to 100 times greater than those of silicone elastomers formed from widely used SPCs such as Sylgard 184 and was comparable to or higher than those of conventional PDMS composites ($\Gamma \sim 10^3$–$10^4$ J m$^{-2}$)[35], despite our $S_{E(25k,25k)}$ did not contain any filler. Another intriguing property is its highly stretchy nature. While several studies indicate that the $\varepsilon_B$ of silicone elastomers formed from Sylgard 184 reach 200%[36,37], $S_{E(25k,25k)}$ showed $\varepsilon_B$ of 330%, demonstrating its highly stretchy nature. A more uniform network from the SPC would avoid stress concentrations at junction points, which likely resulted in the formation of highly stretchy elastomers with excellent mechanical properties.

## Conclusions

In conclusion, we have established a facile, versatile, and scalable methodology for synthesizing polysiloxanes based on the BOROP of D3. Pairing urea with either TBD or a highly basic proton sponge enables the precision synthesis of PDMS with a controlled MW and narrow dispersity, even at

### Table 3 | ROP of D3 initiated from I3 catalyzed by various organic bases with U(4CF3)[a]

| Entry | Cat. | [I3]0 : [Cat.]0 : [HBA]0 : [M]0 | Time (min) | Conv. (%)[b] | $M_n$[c] | $Đ$[d] |
|---|---|---|---|---|---|---|
| 1 | DMAP | 1:1.5:3:135 | 120 | N.D. | | |
| 2 | DMAN | 1:1.5:3:135 | 120 | N.D. | | |
| 3 | TMGN | 1:1.5:3:135 | 120 | 26 | 6700 | 1.06 |
| 4 | TMGN | 1:1.5:3:135 | 360 | 80 | 13,000 | 1.10 |
| 5[e] | TMGN | 1:1.5:3:135 | 360 | N.D. | | |

[a]The polymerizations were performed at 30 °C. [M]0 = 2.4 M. Calculated number average molecular weight at the full consumption of D3 was 30,000.
[b]Conversion determined by $^1$H NMR.
[c]Number average molecular weight, determined by SEC with RI detector.
[d]Dispersity ($Đ = M_w/M_n$), determined by SEC. SEC measurements were performed after terminating with benzoic acid.
[e]U(Cy) was used instead of U(4CF3).

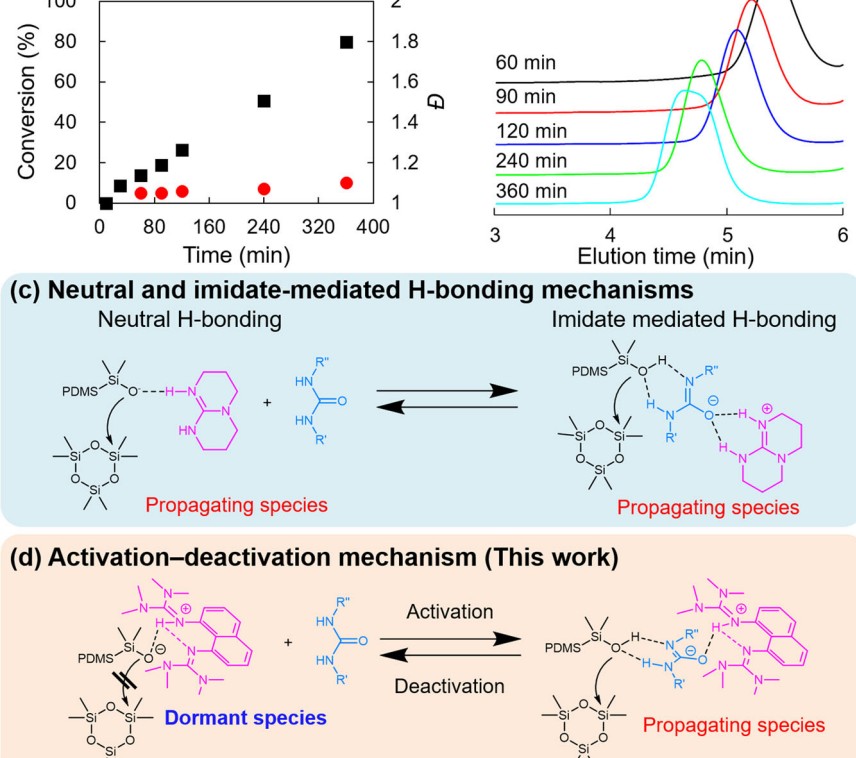

Fig. 4 | Binary organocatalytic ROP for the synthesis of star-shaped PDMSs. a Plots of conversion and $Đ$ against time and (b) SEC traces of the products after a series of polymerization time (60, 90, 120, 240, 360 min) (Table 3, entry 4). c Neutral and imidate-mediated H-bonding mechanisms for base and urea cocatalyzed ROP reported in the literature[26]. d Proposed activation–deactivation mechanism for the present synergistic BOROP catalysis with TMGN and urea. Source data for Fig. 4a is available (Supplementary Data 1).

(a)

(b)

(c) Neutral and imidate-mediated H-bonding mechanisms
Neutral H-bonding

Imidate mediated H-bonding

Propagating species

Propagating species

(d) Activation–deactivation mechanism (This work)

Activation

Deactivation

Dormant species

Propagating species

**Fig. 5 | Formation of silicone elastomers from star-shaped PDMSs synthesized by the present BOROP. a** Fabrication of silicone elastomer (S_E) by hydrosilylation reaction between P_H and P_V. **b** Time-course plots of $G'$ and $G''$ during the formation of S_E (black line, S_E(10k, 10k); blue line, S_E(10k, VS-7); red line, S_E(25k, 25k); purple dashed line, S_E(27k, 10k); green dashed line, S_E(27k, VS-7)). **c** Stress–strain curve of S_E(25k, 25k).

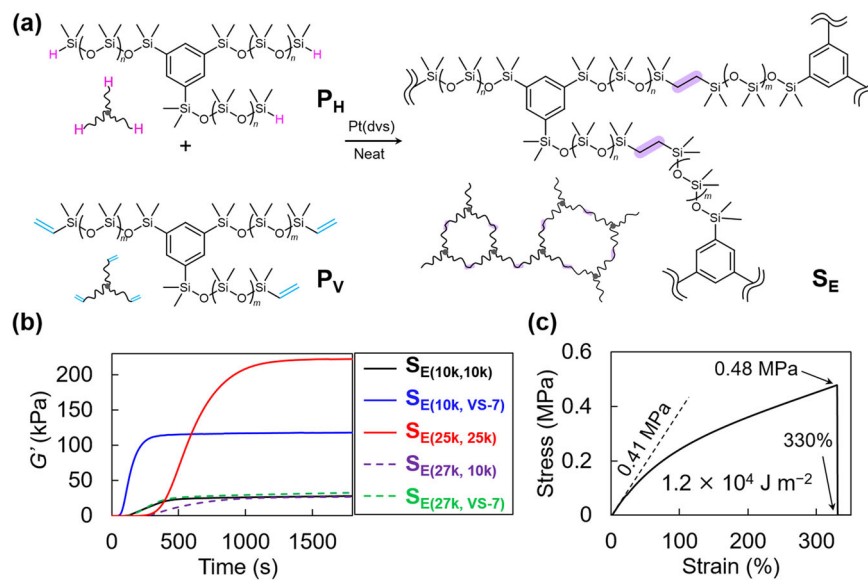

high monomer conversion. The BOROP developed here is also expected to benefit the development of various silicone-based materials after further studies on the effects of solvents[38] and other factors. To the best of our knowledge, the control of ROP using activation–deactivation equilibrium based on the properties of proton sponges is conceptually new. Moreover, star-shaped PDMSs synthesized by the established BOROP system were applied to SPCs, demonstrating the formation of highly stretchy elastomers with excellent mechanical properties. The binary catalyst system developed here has the potential to be applied to various ring monomers and will open the door for applications requiring precision polymer synthesis, such as human-body-related applications.

## Methods
### Materials
D3 (98%, Aldrich), DMAP, DMAN, TBD (98%, Aldrich), and TMGN were first dissolved in superdehydrated stabilizer-free THF (99.5%, Wako) and then dried over MS4A for 24 h before use. Chlorodimethylsilane (98%, Aldrich), chlorodimethylvinylsilane (98%, Aldrich), platinum(0)−1,3-divinyltetramethyldisiloxane complex solution (Karstedt's catalyst) (Pt~2% in xylene, Aldrich) and other reagents were used as received. Industrial-grade PDMS with vinyl end groups and a broad Đ synthesized by equilibrium polymerization, such that the relationship between the branching point and the number of end groups is equivalent to that of 3-armed star-shaped PDMS (VS-7: $M_n$ = 12,000, Lot no. 309001), was generously supplied by Shin-Etsu Chemical Co., Ltd. Benzene-1,3,5-triyl-tris(dimethylsilanol) (**I₃**)[20] and (thio)ureas[9] were synthesized according to previously reported procedures.

### Unitary catalytic ring-opening polymerization
In a typical procedure, a mixture of **I₃** (10 mg, 0.10 mmol for OH groups), D3 (1.0 g, 4.5 mmol), and THF (1.8 mL) was prepared in a vial. To this mixture, a THF solution (0.3 mL) of TBD (7 mg, 0.050 mmol) was added to initiate ROP and the resulting solution was stirred at room temperature (25 °C). Aliquots were taken from the mixture at given times and subjected to ¹H NMR and SEC analyses to determine monomer conversion, $M_n$, and Đ. Likewise, other bases (DMAP, DMAN, and TMGN) were employed.

### Binary catalytic ring-opening polymerization
In a typical procedure, a mixture of U(Cy) (22 mg, 0.10 mmol), **I₃** (10 mg, 0.10 mmol for OH groups), D3 (1.0 g, 4.5 mmol), and THF (1.8 mL) was prepared in a vial. To this mixture, a THF solution (0.3 mL) of TBD (7 mg,

0.050 mmol) was added to initiate ROP and the resulting solution was stirred at room temperature (25 °C). Aliquots were taken from the mixture at given times and subjected to ¹H NMR and SEC analyses to determine monomer conversion, $M_n$, and Đ. Likewise, other combinations of bases (DMAP, DMAN, TBD, and TMGN) and (thio)urea were employed.

### Synthesis of P_H and P_V
In a typical procedure, a mixture of U(Cy) (654 mg, 3.0 mmol), **I₃** (300 mg, 3.0 mmol for OH groups), D3 (30.0 g, 135 mmol), and THF (54 mL) was prepared in a flask. To this mixture, a THF solution (9 mL) of TBD (209 mg, 1.5 mmol) was added to initiate ROP, and the resulting solution was stirred at room temperature (25 °C). After 1 h, a THF solution (9 mL) of benzoic acid (3.7 g, 30 mmol) was added to terminate the polymerization, and the solution was stirred at room temperature (25 °C). After 3 h, the mixture was concentrated to dryness and washed with acetone to afford three-armed star-shaped PDMS with hydroxy end groups. The yield was 25.0 g, which was used for the following end-capping reaction without further purification. Thus, the resulting PDMS (20.0 g) was dissolved in THF (40 mL), pyridine (11.6 mL, 144 mmol), and chlorodimethylsilane (5.23 mL, 48 mmol) in this order, and the resulting mixture was stirred at room temperature. After 17 h, the mixture was poured into excess H₂O/hexane, and the aqueous layer was separated and extracted with hexane. The combined hexane phase was washed with water, dried over Na₂SO₄, and concentrated to dryness. The residue was washed successively with MeOH and acetone and dried under reduced pressure to afford three-armed star-shaped PDMS with hydrosilane end groups (**P_H(25k)**) as a colorless oil. The yield was 16.7 g. ¹H NMR (500 MHz, acetone-$d_6$, $\delta$): 0.07 (–SiO(C$H_3$)₂–), 4.71 (–Si(CH₃)₂$H$), 7.76 (s, Ar$H$). Number-averaged MW ($M_n$) and dispersity ($Đ = M_w/M_n$) measured by SEC calibrated with polystyrene standards; $M_n$ = 25000, Đ = 1.16.

Three-armed star-shaped PDMS with vinyl end groups (**P_V(25k)**) was synthesized from the remaining three-armed star-shaped PDMS with hydroxy end groups (5.0 g) by using chlorodimethylvinylsilane as an end-capping reagent instead of chlorodimethylsilane. The yield was 4.0 g. ¹H NMR (500 MHz, acetone-$d_6$, $\delta$): 0.07 (–SiO(C$H_3$)₂–), 5.72 (dd, –Si(CH₃)₂CH = C$H_2$), 5.92 (dd, –Si(CH₃)₂CH = C$H_2$), 6.13 (dd, –Si(CH₃)₂C$H$ = CH₂), 7.76 (s, Ar$H$). $M_n$ and Đ measured by SEC calibrated with polystyrene standards; $M_n$ = 25000, Đ = 1.15.

Likewise, other 3-armed star-shaped PDMSs with hydrosilane and vinyl end groups (**P_H(10k)**, **P_H(27k)**, and **P_V(10k)**) were synthesized. For ¹H NMR spectra of **I₃**, **P_H(27k)**, and **P_V(25k)**, see Supplementary Data 2.

## Elastomer formation

In a typical procedure, a weighed amount of $P_{V(25k)}$ (1.5 g) and $P_{H(25k)}$ (1.5 g) was first mixed in a vial. To this mixture, Karstedt's catalyst (10 μL) was added to allow the formation of elastomer, and then the mixture was immediately poured into a polystyrene antistatic weighing dish. After 1 h, the solidified PDMS elastomer ($S_E$) with a thickness of 1 mm was peeled from the dish and cut with a dumbbell blade (No. 7) to form a dumbbell-shaped tensile test specimen (ISO 527-2:2012, Type 5B). A tensile test of the resulting specimen was conducted at least 2 h after specimen preparation. For rheological measurements, the elastomer formation was performed on a measurement stage of an Anton Paar MCR 102 rheometer.

## Data availability

All data supporting the findings of this study are available within this article and its Supplementary Information file. Source Data for Figs. 2b, 3b, and 4a can be found in Supplementary Data 1. The Original $^1$H NMR spectra of the compounds obtained in this manuscript are available in Supplementary Data 2. The data are also available from the corresponding author upon reasonable request.

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

## Acknowledgements
The authors are grateful to Dr. Minami Oka (The University of Tokyo) and Mr. Kazuki Fuke (The University of Tokyo) for their kind support for the synthesis of PDMSs. This work was supported by JSPS KAKENHI (Grant Numbers 21H01632 and 23K17337 S.H.), Fuji Seal Foundation (S.H.), and UTEC-UTokyo FSI Research Grant Program (S.H.).

## Author contributions
S.H. conceived the concept of the project. H.O. performed the polymer synthesis and characterization. H.O., A.S and S.H. performed elastomer formation and mechanical tests. S.H. wrote the paper, and S.H. supervised the project. All authors discussed the manuscript.

## Competing interests
S.H. is an Editorial Board Member for *Communications Chemistry* but was not involved in the editorial review of, or the decision to publish this article. The authors declare no competing interests.
