## [Peer Review File · Communications Chemistry]

Reviewers' comments:

Reviewer #1 (Remarks to the Author):

This is a timely and useful paper that should be published. With the recent publication of numerous reports of “bottle brush” and other topology silicone networks using ill defined chemistries, this paper will be welcomed.

I am fine with the paper being published “as is”, but the authors might consider tempering their “synergetic binary organocatalytic” arguments after reading the 2 papers referenced below. The success of these syntheses involves the competition between propagation and back-biting and their suggestions might be merely solvent effects. See:

Goff, J.; Sulaiman, S.; Arkles, B. Applications of Hybrid Polymers Generated from Living Anionic Ring Opening Polymerization. *Molecules* 2021, 26, 2755. (this paper should be referenced)

Fessler, W.A.; Juliano, P.C. Reactivity of Solvated Lithium N-Butyldimethylsilanolate with Organosiloxane Substrates. *Ind. Eng. Chem. Prod. Res. Dev.* 1972, 11, 407–410.

Reviewer #2 (Remarks to the Author):

The manuscript describes a method for making star-shaped polysiloxanes by ring-opening polymerization of D3 using benzene-1,3,5-triyltris(dimethylsilanol).

Narrow molecular weight distributions of the star-shaped polysiloxanes are obtained by combining a superbase and a urea as a binary catalytic system. The polymers were crosslinked after derivatization to give silicon elastomers which were characterized by rheological methods.

I do not recommend publishing the manuscript in its present form and content mainly because of the following reasons:

- 1) The state of the art is not cited properly. Statements are not sufficiently supported by literature citations, sometimes unclear or blurred.
- 2) The analytical methods applied are inadequate for the interpretation of the results. Experimental and analytical data are missing which would be necessary for the interpretation of the results.

To 1)-Examples

Line 16-19 (abstract): These statements should be supported in the main section by the literature.

Lines 40-41: generalization is incorrect, regard e.g. phosphazenes (probably toxic), Fe or Mn complexes (nontoxic), NaOH (nontoxic) etc.

Line 44,45: is incorrect, “historic” are the given references. How are polysiloxanes made today? Please cite adequately.

Line 48: ref. 14 is not cited correctly. Ref. 14 describes a method to make uniform polysiloxanes very simply by anionic living polymerization, please discuss. See also JP04331233.

Line 49 and 54 / Lines 93-95: A 5-membered ring carbosilane is used in ref. 15, 16, not a cyclic siloxane.

Please cite references for the ROP of D3 appropriately.

Important is e.g.: Fuchise, K.; Kobayashi, T.; Sato, K.; Igarashi, M. Organocatalytic ring-opening polymerization of cyclotrisiloxanes using silanols as initiators for the precise synthesis of asymmetric linear polysiloxanes. *Polymer Chem.* 2020, 11, 7625–7636. Furthermore e.g. Despotopoulou, C.; Klein, J.; Hemery, T.; Koehler, T.; Grefe, L.; Mejia, E.; Kragl, U. Process for the preparation of polyorganosiloxanes. Patent EP3599256 A1.

Line 66: “remains a crucial challenge”: please discuss in the light of ref. 14, where the problems have been solved using lithium salts.

Lines 67-82: contain a summary of results/statements – please shift to the main part.

Lines 85: catalytic activity - in which reactions? How? Please cite.

Lines 101-105, Table 1, Lines 135 ff: The well-known bifunctionality of TBD, that is the capability to take up and release protons at the same time, should be mentioned here, see e.g. E. Fritz-Langhals, *Org. Proc. Res. Dev.* 2022 and references cited herein.

Line 162: How can this unusual linearity be explained?

Page 13, Fig. 4c,d: There is no experimental evidence of the hypotheses on page 13. IR spectroscopy is recommended here to elucidate the existence of compounds as in Fig. 4 with multiple different hydrogen bonds.

Fig. 4 c: Literature is not given, please cite.

Lines 190 ff: Are the mechanical properties of the elastomer due to the narrow distribution of molecular weight of the siloxanes? They could also be the result of the star-shaped monomers. Did you check the degree of crosslinking?

To 2) - Examples:

The conversion of the polymerizations was followed by ¹H NMR. This is inadequate, because the amount of D3 cannot be determined in the mixture. Which signals could be used then? ²⁹Si NMR analysis to determine educt and product species is strongly recommended here. Catalyst deactivation should be performed in all experiments to stop the reaction immediately.

The statements of page 9 are not supported experimentally. The ¹H NMR spectra discussed on page 9 are very noisy. The very small signals are not integrated. The assignment of the peaks is not clear.

Effective hydrogen bonding is not proved.

Table 1 - 3: Molar excesses of catalyst / cocatalyst are used. Is there a positive effect with “catalytical amounts”, say 0,1 mol %? That would be interesting on preparative reasons.

Fig. 2 c and 3 c: There are two maxima formed in the SEC spectra during polymerization. This inevitably needs clarification. Are both peaks used for the calculation of D? Small differences in D, Table 3 and 4, are not significant and should not be discussed. Otherwise the experiments should be reproduced.

Line 122: “became turbid due to the insolubility...” how is that proved?

p. 16: There is no proof that the mechanical properties found are due to a narrow molecular weight distribution of the polymer but to the special topology and composition of the polymer. Comparison using the same type of polymer with a broader MW distribution is necessary.

The key compound I3 contains impurities in larger amounts, as can be seen in the ¹H NMR spectrum.

Have these impurities been identified? They might affect stoichiometry and reactivity.

Lines 280 ff. How was the degree of crosslinking? => This influences the mechanical properties strongly and should be determined, e.g. spectroscopically.

Further remarks

The abbreviation PDMS stands for linear polydimethylsiloxane, please specify: e.g. star-shaped PDMS etc., all abbreviations have to be explained at their first appearance.

Wording is sometimes imprecise: e.g. Line 57: multifunctional => trifunctional silanol + precise name

Minor mistakes

Fig.2 Formula of P is wrong – there are 3 Si-OH end-groups

line 13: versatile

line 34: narrow dispersities

Line 107: unitary

After all, language polishing will help to improve readability and may also contribute to improve clarity.

Response to Referees

First, we thank the editor and reviewers very much for their valuable comments and suggestions. **We all strongly support a transparent peer review process and are very pleased that this file will be published as a Peer Review File in one of the supplementary information files upon acceptance.**

We have thoroughly reviewed all of the comments raised by the reviewers and have elaborated the manuscript. Texts revised according to the comments have been indicated in red.

Reviewer #1:

This is a timely and useful paper that should be published. With the recent publication of numerous reports of “bottle brush” and other topology silicone networks using ill defined chemistries, this paper will be welcomed.

I am fine with the paper being published “as is”, but the authors might consider tempering their “synergetic binary organocatalytic” arguments after reading the 2 papers referenced below. The success of these syntheses involves the competition between propagation and back-biting and their suggestions might be merely solvent effects. See:

Goff, J.; Sulaiman, S.; Arkles, B. Applications of Hybrid Polymers Generated from Living Anionic Ring Opening Polymerization. *Molecules* 2021, 26, 2755. (this paper should be referenced)

Fessler, W.A.; Juliano, P.C. Reactivity of Solvated Lithium N-Butyldimethylsilanolate with Organosiloxane Substrates. *Ind. Eng. Chem. Prod. Res. Dev.* 1972, 11, 407–410.

We are grateful to Reviewer #1. We were greatly encouraged by these comments. The suggested literature and a recent study by Shi et al. (*Science* 381, 1011–1014 (2023)) have been cited. The relevant discussion has accordingly been updated in the revised manuscript.

Reviewer #2:

The manuscript describes a method for making star-shaped polysiloxanes by ring-opening polymerization of D3 using benzene-1,3,5-triyltris(dimethylsilanol). Narrow molecular weight distributions of the star-shaped polysiloxanes are obtained by combining a superbases and a urea as a binary catalytic system. The polymers were crosslinked after derivatization to give silicon elastomers which were characterized by rheological methods.

I do not recommend publishing the manuscript in its present form and content mainly because of the following reasons:

- 1) The state of the art is not cited properly. Statements are not sufficiently supported by literature citations, sometimes unclear or blurred.
- 2) The analytical methods applied are inadequate for the interpretation of the results. Experimental and analytical data are missing which would be necessary for the interpretation of the results.

We deeply thank Reviewer #2 for the valuable comments based on deep insights into the various prior literature and even into Japanese domestic patent applications. We have thoroughly reviewed all the comments, but we respectfully disagree with some of the comments as responded in detail as follows. Nevertheless, we believe that the significance of our manuscript has been greatly improved by revisions based on the remaining comments.

To 1)-Examples

Line 16-19 (abstract): These statements should be supported in the main section by the literature.

These statements have been supported by literatures in the Introduction section.

Lines 40-41: generalization is incorrect, regard e.g. phosphazenes (probably toxic), Fe or Mn complexes (nontoxic), NaOH (nontoxic) etc.

The notion that organocatalysts are less toxic than metal complex catalysts has frequently been explained. For recent reviews, see:

- Amaury Bossion, Katherine V. Heifferon, Leire Meabe, Nicolas Zivic, Daniel Taton, James L. Hedrick, Timothy E. Long, Haritz Sardon, *Prog. Polym. Sci.*, 90, 164–210.
- Winnie Nzahou Ottou, Haritz Sardon, David Mecerreyes, Joan Vignolle, Daniel Taton, *Prog. Polym. Sci.*, 56, 64–115.

The counterexample to this raised by the reviewer was ‘phosphazenes (probably toxic)’. Of course, the reviewer would understand that one should not judge the toxicity of materials by presumption, but the toxicity concerns with ‘phosphazenes’ are understandable. This made us realize that there is room for improvement in the text on toxicity. Hence, we have rephrased the relevant text to avoid misleading readers (lines 40-41) and the above papers have been cited in the updated version of our manuscript.

Line 44,45: is incorrect, “historic” are the given references. How are polysiloxanes made

today? Please cite adequately.

In the previous version, we cited the literature by Prof. Waymouth and Dr. Hedrick, who have established the field of organocatalytic ROP itself and developed the synthesis of silicon-containing polymers including PDMS based on organocatalytic ROP (e.g., *Org. Lett.* **8**, 4683-4686, 2006). In addition, we have cited our extension of this chemistry (Minami Oka, Hideaki Takagi, Tomotaka Miyazawa, Robert M. Waymouth, and Satoshi Honda, *Adv. Sci.* 2021, **8**, 2101143). Of course, these are the syntheses of polysiloxanes “today” by organocatalytic ROP. Moreover, we have cited a very recent example of synthesis of polysiloxanes reported in *Science* (Shi et al., *Science* **381**, 1011–1014 (2023)) and the suggested literature by Reviewer #1 in the updated version.

Line 48: ref. 14 is not cited correctly. Ref. 14 describes a method to make uniform polysiloxanes very simply by anionic living polymerization, please discuss. See also JP04331233.

We appreciate the reviewer's comments, not only about the information in the research paper, but also about his or her deep understanding of domestic patents written only in Japanese.

Regarding the texts for introducing ref 14, which describes **non-organocatalytic ROP**, perhaps, the wording ‘meticulous care’ might be unclear. In the updated version, we have rephrased the text as ‘...applications require a controlled molecular weight (MW) with a narrow \mathcal{D} ’. Related texts have also been updated.

We are aware of the importance of JP04331233, which is one of the same day domestic patent applications by the same inventors written in Japanese (patent application: JP04331233 and JP04331234) as **non-organocatalytic ROP**. Although no GPC curves are presented in the literature and only numerical information is given, the narrowest distribution of PDMS written in the literature was $\mathcal{D} = 1.13$ ($M_n = 12000$). Other reported PDMSs had relatively broad \mathcal{D} s for which we observed broadening or splitting of the GPC curve in our present study. **Although such discussion would be possible, it will not allow non-Japanese readers to trace it back to the source literature. Moreover, in the reviewer’s words, these would be classified as ‘historic’ ROP (non-organocatalytic ROP) as in the previous comment.**

Line 49 and 54 / Lines 93-95: A 5-membered ring carbosilane is used in ref. 15, 16, not a cyclic siloxane.

Regarding ref. 15 in the previous version (Waymouth et al., *Org. Lett.* **8**, 4683-4686 (2006)), although the graphical abstract and title say so, it will be immediately apparent

from Scheme 4 and the extremely generous supporting information that organocatalytic ROP of cyclic siloxane was also carried out. It is hard to believe that this reviewer, who is extremely knowledgeable in this field, would make such an oversight.

Please cite references for the ROP of D3 appropriately.

Important is e.g.: Fuchise, K.; Kobayashi, T.; Sato, K.; Igarashi, M. Organocatalytic ring-opening polymerization of cyclotrisiloxanes using silanols as initiators for the precise synthesis of asymmetric linear polysiloxanes. *Polymer Chem.* 2020, 11, 7625–7636. Furthermore e.g. Despotopoulou, C.; Klein, J.; Hemery, T.; Koehler, T.; Grefe, L.; Mejia, E.; Kragl, U. Process for the preparation of polyorganosiloxanes. Patent EP3599256 A1.

As with the previous comment, the comments regarding the recent synthesis of silicon-containing polymers by organocatalyzed ROP by this reviewer, who must be deeply familiar with the field, may be based on some misunderstanding. Otherwise, there seems to be something inconvenient for this reviewer that the ROP of D3 was reported quite a few years ago.

For the use of silanol as an initiator, an appropriate paper has already been cited (Waymouth et al., *Org. Lett.* 8, 4683-4686 (2006)). In this report, the synthesis of silicon-containing polymers including polysiloxanes by organocatalytic ROP with guanidinium bases such as TBD and N-heterocyclic carbenes (NHCs), and the use of silanol as an initiator for organocatalytic ROP have been reported. It would be sufficient to cite this, since the suggested journal article relies on the same chemistry but published more than 10 years later without citing the above *Org. Lett.* paper and related pioneering works.

For the suggested patent application, we are grateful to the reviewer for this information. In this patent application, 8 examples and 2 comparative examples are described, 9 of which used TBD as a catalyst for ROP. In Example 5, 1,8-bis(tetramethylguanidino)naphthalene (TMGN) was used for ROP of D4 at 65 °C, and the resulting small molecular weight product ($M_n = 3000$) indicates the progress of equilibrium polymerization. As is clear from the overall content, this patent application describes only the unitary-organocatalytic ROP using these bases and did not envision binary-organocatalytic ROP (BOROP). This patent application would strongly support the novelty and significance of our manuscript. The related discussion has been updated.

Line 66: “remains a crucial challenge”: please discuss in the light of ref. 14, where the problems have been solved using lithium salts.

The ROP described in ref 14 in the previous version did not use a trifunctional initiator

and is not a paper related to organocatalytic ROP. As also described in the previous version, the trifunctional silanol (I3) had poor solubility even when not in a salt form, and our research conceptually propose that the methodology for dissolving initiators irrespective of counter cationic species.

Lines 67-82: contain a summary of results/statements – please shift to the main part.

As many journals have similar guidelines, Communications Chemistry also recommends that we write a brief summary of the major results and conclusions. The reviewer should read through ‘style and formatting guide’ of the journal.

Lines 85: catalytic activity - in which reactions? How ? Please cite.

The text was rephrased using ‘basicity’. Citations for pK_{BH^+} have already been presented.

Lines 101-105, Table1, Lines 135 ff: The well-known bifunctionality of TBD, that is the capability to take up and release protons at the same time, should be mentioned here, see e.g. E. Fritz-Langhals, Org. Proc. Res. Dev. 2022 and references cited herein.

We thank the reviewer for the suggestion. The suggested literature has been cited in the updated manuscript.

Line 162: How can this unusual linearity be explained?

The linearity of the results was explained with the proposed activation-deactivation mechanism, as discussed in the previous version. Similar to various organic reactions for which detailed mechanisms have not been elucidated, it is difficult to elucidate the detailed mechanism. Nevertheless, for the clarity, the proposed mechanism was further discussed based on infrared (IR) spectroscopy in the present version of our manuscript.

Page 13, Fig. 4c,d: There is no experimental evidence of the hypotheses on page 13. IR spectroscopy is recommended here to elucidate the existence of compounds as in Fig. 4 with multiple different hydrogen bonds.

We appreciate the suggestion. IR spectra have been presented in the supplementary information to support our hypothesis.

Fig. 4 c: Literature is not given, please cite.

In the updated version, the relevant paper has been cited.

Lines 190 ff: Are the mechanical properties of the elastomer due to the narrow distribution

of molecular weight of the siloxanes? They could also be the result of the star-shaped monomers. Did you check the degree of crosslinking?

To experimentally confirm the dependence of \mathcal{D} on mechanical properties, we compared the mechanical properties of elastomers synthesized from a series of combinations of star-shaped PDMSs with SiH and vinyl end groups in the updated version of our manuscript. The degree of crosslinking was also determined based on IR spectroscopy (Supplementary Figures 4 and 5).

To 2) - Examples:

The conversion of the polymerizations was followed by ^1H NMR. This is inadequate, because the amount of D3 cannot be determined in the mixture. Which signals could be used then? ^{29}Si NMR analysis to determine educt and product species is strongly recommended here. Catalyst deactivation should be performed in all experiments to stop the reaction immediately.

Determining the conversion of D3 by ^1H NMR is adequate and routine and can be found elsewhere. An example of an organocatalytic ROP in which conversion is evaluated by ^1H NMR has already been cited (Waymouth et al., *Org. Lett.* **8**, 4683-4686 (2006)), where the spectrometer was operated at 400 MHz. As described in the experimental section, we operated our NMR spectrometer at 500 MHz, and, of course, monitoring the conversion of D3 is much easier.

To prepare time-conversion plots, quick dilution of the polymerization mixture with deuterated solvent was enough to avoid further polymerization or scrambling reactions. We, of course, performed catalyst deactivation with benzoic acid for large-scale synthesis as described in the 'Synthesis of P_H and P_V ' section in the experimental part.

The statements of page 9 are not supported experimentally. The ^1H NMR spectra discussed on page 9 are very noisy. The very small signals are not integrated. The assignment of the peaks is not clear. Effective hydrogen bonding is not proved.

As described in the previous version of our manuscript, we used 80 MHz benchtop NMR, which allows ^1H NMR measurements at concentrations comparable to those used for polymerization. Low resolution and noise are unavoidable due to the inherent performance of the benchtop NMR systems. Of course, we cannot estimate the degree of hydrogen bonding in the measurement solution with the concentration comparable to polymerization reaction mixtures by using a large high-resolution NMR instrument due to the mismatch between the concentration of the polymerization solution and the proper concentration for ^1H NMR. In contrast, the benchtop system allows the analysis of the

effect of hydrogen bonding on signals in THF at the concentration comparable to those of the polymerization mixtures. It is common practice to use different devices depending on what is being discussed. Also, it is apparent that the measurements were not intended for assignment.

On the other hand, as was pointed in the previous comment, hydrogen bonding interactions were further evaluated based on IR spectroscopy in the updated version of our manuscript.

Table 1 - 3: Molar excesses of catalyst / cocatalyst are used. Is there a positive effect with “catalytical amounts”, say 0,1 mol %? That would be interesting on preparative reasons. Fig.2 c and 3 c: There are two maxima formed in the SEC spectra during polymerization. This inevitably needs clarification. Are both peaks used for the calculation of D ? Small differences in D , Table 3 and 4, are not significant and should not be discussed. Otherwise the experiments should be reproduced.

We confirmed that it is possible to reduce the amount of catalyst to 1 mol% but we did not adopt this condition because of the extremely slow polymerization rate.

SEC is not a spectrum but a chromatogram. We have analyzed two peaks, and numerical information is presented in the updated version of our manuscript.

We discussed the D s not only by numerical information but also by the form of chromatograms. We believe that the spread of the distribution on the chromatograms is clear and arguable even when the numerical difference is small.

Line 122: “became turbid due to the insolubility...” how is that proved?

It is apparent that if we observe the reaction solution. We were able to clearly observe that the solution became turbid.

p. 16: There is no proof that the mechanical properties found are due to a narrow molecular weight distribution of the polymer but to the special topology and composition of the polymer. Comparison using the same type of polymer with a broader MW distribution is necessary.

To experimentally confirm the dependence of D on mechanical properties, we compared the mechanical properties of elastomers synthesized from a series of combinations of star-shaped PDMSs with SiH and vinyl groups.

The key compound I3 contains impurities in larger amounts, as can be seen in the ¹H NMR spectrum. Have these impurities been identified? They might affect stoichiometry and reactivity.

The spectrum is presented in the supplementary material. The deuterated solvent used for the ¹H NMR measurement of I3 was (CD₃)₂CO, and the signals at 2.05 and 2.81 ppm, and 2.84 ppm are acetone-, HDO-, and H₂O-derived signals (solvent residual signals), respectively. We thought these were very common and thus did not need to show them on the spectrum. However, in the updated version, we have indicated these on the spectrum for the benefit of readers unfamiliar with ¹H NMR characterization.

Lines 280 ff. How was the degree of crosslinking ? => This influences the mechanical properties strongly and should be determined, e.g. spectroscopically.

In the updated version, the degree of crosslinking was determined based on IR spectroscopy.

Further remarks

The abbreviation PDMS stands for linear polydimethylsiloxane, please specify: e.g. star-shaped PDMS etc., all abbreviations have to be explained at their first appearance.

Wording is sometimes imprecise: e.g. Line 57: multifunctional => trifunctional silanol + precise name

Minor mistakes

Fig.2 Formula of P is wrong – there are 3 Si-OH end-groups

line 13: versatile

line 34: narrow dispersities

Line 107: unitary

After all, language polishing will help to improve readability and may also contribute to improve clarity.

Regarding wording, we have thoroughly reviewed our manuscript again and have updated if needed. The abbreviation PDMS apparently stands for poly(dimethyl siloxane), and the term PDMS does not contain ‘linear’ and any type of macromolecular architectures. As is clear from the IUPAC recommendation, where ‘star-polyisoprene’ is given as an example, polymer class names, including macromolecular architectures, precede before the abbreviated names of polymers.

On the other hand, of course, the star-shaped PDMSs synthesized in the present study have three OH groups. But, why does this reviewer think the formula of P is wrong? Again, based on the IUPAC recommendation (DOI: 10.1039/9781847559425), the rule

only says 'The formulae of end groups, if known, may be attached to the bonds at the ends of the constitutional units, but placed outside the enclosing marks'. Of course, we frequently see the omitted forms of the formulae of end groups if they are self-evident. In the present case, it is obvious that the end groups are three OH groups as also pointed out by the reviewer. However, given the journal's wide readership, the end groups have been added for the benefit of readers unfamiliar with polymer chemistry.

We have updated our manuscript based on other remaining comments if needed.

Reviewer #2:

There are only two main issues left, please see below.

Black: first review

Red: Author's response

Black: 2nd review

The key compound I3 contains impurities in larger amounts, as can be seen in the ¹H NMR spectrum. Have these impurities been identified? They might affect stoichiometry and reactivity.

The spectrum is presented in the supplementary material. The deuterated solvent used for the ¹H NMR measurement of I3 was (CD₃)₂CO, and the signals at 2.05 and 2.81 ppm, and 2.84 ppm are acetone-, HDO-, and H₂O-derived signals (solvent residual signals), respectively. We thought these were very common and thus did not need to show them on the spectrum. However, in the updated version, we have indicated these on the spectrum for the benefit of readers unfamiliar with ¹H NMR characterization.

The interpretation of the spectrum of I3 is anything but obvious (water has a variable chemical shift in the NMR spectrum which depends on the composition of the solution). You used wet deuterated acetone, alright, but why should you have HDO ?? Furthermore, if d₆ acetone should exchange with H₂O, then Si-OH should also exchange with H₂O. Nevertheless, CDCl₃ should be used as solvent, because you have also used CDCl₃ for the product mixture. As you might know, the chemical shifts vary considerably with the solvent.

Now, with the high resolution NMR spectrum of the product mixture you are able to determine the conversion! Unfortunately, you have not separated the integrals for D3 and the polymer which is necessary to do so. Please take a closer look at the cited Waymouth paper. Waymouth also determines the conversion from the signals of the polymer and the signals of D3. Please, assign these signals in your NMR spectrum.

Please refer then to the received value of conversion in the main part of the manuscript.

Further remarks

The abbreviation PDMS stands for linear polydimethylsiloxane, please specify: e.g. star-shaped PDMS etc., all abbreviations have to be explained at their first appearance.

Wording is sometimes imprecise: e.g. Line 57: multifunctional => trifunctional silanol + precise name

Minor mistakes

Fig.2 Formula of P is wrong – there are 3 Si-OH end-groups

line 13: versatile

line 34: narrow dispersities

Line 107: unitary

After all, language polishing will help to improve readability and may also contribute to improve clarity.

Regarding wording, we have thoroughly reviewed our manuscript again and have updated if needed. The abbreviation PDMS apparently stands for poly(dimethyl siloxane), and the term PDMS does not contain 'linear' and any type of macromolecular architectures. As is clear from the IUPAC recommendation, where 'star-polyisoprene' is given as an example, polymer class names, including macromolecular architectures, precede before the abbreviated names of polymers.

PDMS contains only dimethylsiloxy groups, o.k?

On the other hand, of course, the star-shaped PDMSs synthesized in the present study have three OH groups. But, why does this reviewer think the formula of P is wrong? Again, based on the IUPAC recommendation (DOI: 10.1039/9781847559425), the rule only says 'The formulae of end groups, if known, may be attached to the bonds at the ends of the constitutional units, but placed outside the enclosing marks'. Of course, we frequently see the omitted forms of the formulae of end groups if they are self-evident. In the present case, it is obvious ?? that the end groups are three OH groups as also pointed out by the reviewer. However, given the journal's wide readership, the end groups have been added for the benefit of readers unfamiliar with polymer chemistry.

We have updated our manuscript based on other remaining comments if needed.

I am sorry to say again according to basic knowledge that the formula P is wrong, because it would mean *definitely that the polymer has 3 methyl end groups*. You have to add H at the end of the chain, because you have OH terminal groups. That's simple nomenclature. If you don't believe that, please count the atoms on the left side of your reaction equation and on the right side – left and right side of the equation must be identical! In addition, you have to write "3n" on the left side instead of "n"

Response to Referees

We thank the editor for the patient handling of our manuscript. **We all strongly support a transparent peer review process and are very pleased that this file will be published as a Peer Review File in one of the supplementary information files upon acceptance.**

We also thank Reviewer #2 for the valuable comments in the previous round based on deep insights into the silicon-related chemistry, ROP of D3, and even into domestic patent applications written in Japanese. We are pleased to see that the Reviewer #2 would have accepted most of our response on historical aspects of organocatalytic ROP synthesis of silicone-containing polymers, additional data, and updated Results and discussion. Based on the remaining concerns below, we have further updated our manuscript.

Reviewer #2:

There are only two main issues left, please see below.

Black: first review

Red: Author's response

Black: 2nd review

The key compound I3 contains impurities in larger amounts, as can be seen in the ¹H NMR spectrum. Have these impurities been identified? They might affect stoichiometry and reactivity.

The spectrum is presented in the supplementary material. The deuterated solvent used for the ¹H NMR measurement of I3 was (CD₃)₂CO, and the signals at 2.05 and 2.81 ppm, and 2.84 ppm are acetone-, HDO-, and H₂O-derived signals (solvent residual signals), respectively. We thought these were very common and thus did not need to show them on the spectrum. However, in the updated version, we have indicated these on the spectrum for the benefit of readers unfamiliar with ¹H NMR characterization.

The interpretation of the spectrum of I3 is anything but obvious (water has a variable chemical shift in the NMR spectrum which depends on the composition of the solution). You used wet deuterated acetone, alright, but why should you have HDO ?? Furthermore, if d₆ acetone should exchange with H₂O, then Si-OH should also exchange with H₂O. Nevertheless, CDCl₃ should be used as solvent, because you have also used CDCl₃ for the product mixture. As you might know, the chemical shifts vary considerably with the solvent.

Now, with the high resolution NMR spectrum of the product mixture you are able to determine the conversion! Unfortunately, you have not separated the integrals for D3 and the polymer which is necessary to do so. Please take a closer look at the cited Waymouth paper. Waymouth also determines the conversion from the signals of the polymer and the

signals of D3. Please, assign these signals in your NMR spectrum.

Please refer then to the received value of conversion in the main part of the manuscript.

Response

In the present case, acetone-d₆ was purchased from a manufacturer and used as received without drying, e.g., over molecular sieves. It is an entry-level textbook matter that the HDO-derived signal appears in addition to the H₂O-derived signal in acetone-d₆. In addition to textbooks elsewhere, the existence of H₂O and HDO is mentioned, for example, in a paper described on impurities in deuterated solvents (DOI: 10.1021/acs.oprd.5b00417), the NMR Solvent Data Chart provided by a deuterated solvents manufacturer (Cambridge Isotope Laboratories: <https://cil.showpad.com/share/54zWw3avN5TUYa6uMSY32>), and, of course, in a classic paper describes on how deuterated solvents are produced (DOI: 10.1021/acs.oprd.5b00417). As it is natural for H₂O and HDO to be mixed in because of the method it is produced, it would be rare to call such commercially available one as wet deuterated acetone.

Regarding the determination of conversion, Reviewer #2 commented as ‘*Unfortunately, you have not separated the integrals for D3 and the polymer which is necessary to do so*’. But, where are the ¹H NMR spectra from which we need to separate the integrals? What we presented in the supplementary NMR file is the ¹H NMR of the isolated PDMSs according to the guideline of ‘Provision of NMR spectra’ in the journal.

On the other hand, **we are glad to see that the Reviewer #2 now recognizes that D3 (cyclic siloxane) has been polymerized in the cited reference by the pioneers of organocatalytic ROP; Hedrick and Waymouth (*Org. Lett.* **8**, 4683-4686 (2006)), as he or she pointed out that ‘A 5-membered ring carbosilane is used in ref. 15, 16, not a cyclic siloxane’ in the previous round.** We always take a closer look at ¹H NMR spectra and have prepared time-conversion plots from our previous report with one of the above pioneers (*Adv. Sci.* **8**, 2101143 (2021)). As D3 is one of the very common monomers and ¹H NMR spectra of D3 and PDMS are well-known, the points made by Reviewer #2 regarding ¹H NMR would not be considered meaningful in conveying to the reader the essence of this paper. However, **according to the recommendation from the journal guideline, numerical source data for graphs and charts has been provided as Supplementary Data in Excel format.**

Further remarks

The abbreviation PDMS stands for linear polydimethylsiloxane, please specify: e.g. star-shaped PDMS etc., all abbreviations have to be explained at their first appearance.

Wording is sometimes imprecise: e.g. Line 57: multifunctional => trifunctional silanol + precise name

Minor mistakes

Fig.2 Formula of P is wrong – there are 3 Si-OH end-groups

line 13: versatile

line 34: narrow dispersities

Line 107: unitary After all, language polishing will help to improve readability and may also contribute to improve clarity.

Regarding wording, we have thoroughly reviewed our manuscript again and have updated if needed. The abbreviation PDMS apparently stands for poly(dimethyl siloxane), and the term PDMS does not contain 'linear' and any type of macromolecular architectures. As is clear from the IUPAC recommendation, where 'star-polyisoprene' is given as an example, polymer class names, including macromolecular architectures, precede before the abbreviated names of polymers.

PDMS contains only dimethylsiloxo groups, o.k?

On the other hand, of course, the star-shaped PDMSs synthesized in the present study have three OH groups. But, why does this reviewer think the formula of P is wrong? Again, based on the IUPAC recommendation (DOI: 10.1039/9781847559425), the rule only says 'The formulae of end groups, if known, may be attached to the bonds at the ends of the constitutional units, but placed outside the enclosing marks'. Of course, we frequently see the omitted forms of the formulae of end groups if they are self-evident. In the present case, it is obvious ?? that the end groups are three OH groups as also pointed out by the reviewer. However, given the journal's wide readership, the end groups have been added for the benefit of readers unfamiliar with polymer chemistry.

We have updated our manuscript based on other remaining comments if needed.

I am sorry to say again according to basic knowledge that the formula P is wrong, because it would mean *definitely that the polymer has 3 methyl end groups*. You have to add H at the end of the chain, because you have OH terminal groups. That's simple nomenclature. If you don't believe that, please count the atoms on the left side of your reaction equation and on the right side – left and right side of the equation must be identical! In addition, you have to write "3n" on the left side instead of "n"

Response

As responded in the previous round, we do not believe that the structural formula is wrong,

nor do we believe that any reader would think that all the end groups are methyl groups. In the previous round, this reviewer commented as ‘wrong, incorrect, or inadequate’ on various points, but most of those comments were overturned by our rebuttal. To this comment, we also explained based on the IUPAC recommendations, but they still do not seem to understand. Again, based on the IUPAC recommendations (DOI: 10.1039/9781847559425), the rule only says ‘*The formulae of end groups, if known, may be attached to the bonds at the ends of the constitutional units, but placed outside the enclosing marks*’ (same as the response in the previous round). For example, the polymerization equation that appears at the beginning of the introduction of the IUPAC recommendations also omits the end groups outside of the parentheses (IUPAC recommendations, DOI: 10.1515/pac-2018-0602). The Reviewer #2 might say that the structure in the IUPAC recommendations is wrong and that both end groups are also methyl groups. However, we, and researchers who draw the chemical structures of polymers according to IUPAC recommendations, do not think so. Despite these basis, we thought that we had added ‘OH’ through the first round because the recommendation says ‘*The formulae of end groups, if known, may be attached to the bonds at the ends of the constitutional units, but placed outside the enclosing marks*’ (IUPAC recommendations, DOI: 10.1039/9781847559425). However, it appears that the presented chemical structure was older version likely due to software error or something. In the latest version, we have added OH groups. But this is not because the previous structural formula was wrong. Regarding typo, we have revised them in the updated version of our manuscript.

On the other hand, from the comment ‘*In addition, you have to write “3n” on the left side instead of “n”*’, Reviewer #2 is probably under the misapprehension that ‘n’ must always represent the degree of polymerization. It is also a textbook matter and IUPAC recommendations clearly say that ‘**n**’ represents the ‘**degree of polymerization**’ or ‘**repetition**’ (IUPAC recommendations, DOI: 10.1515/pac-2018-0602). Hence, it is quite common to write ‘n’ outside the parentheses to represent the ‘repetition’ of polymer chains. Of course, we used ‘n’ to represent ‘repetition’.

Furthermore, the number of monomer-derived silicon is already 3n on the left side and 3n on the right side of the polymerization equation, since D3 consists of three repeating units (and the star-shaped PDMSs have three arms), *If you don’t believe that, please count the atoms on the left side of your reaction equation and on the right side.*